# Prevalence and risk factors of early postoperative seizures in patients with glioma: A protocol for meta-analysis and systematic review

Bo Sun[2☉], Wenpeng Lu[2☉], Wangyang Yu[2], Ye Tian[2], Peng Wang [1,2]*

1 Department of Neurosurgery, The Second Affiliated Hospital of Hainan Medical University, Haikou, China,
2 The Second Hospital of Hebei Medical University, Shijiazhuang, Hebei, China

☉ These authors contributed equally to this work.
* 13363893886@163.com

**Data Availability Statement:** No datasets were generated or analysed during the current study. All relevant data from this study will be made available upon study completion.

## Abstract

### Introduction

Early postoperative seizures has been the most common clinical expression in gliomas; however, the incidence and risk factors for early postoperative seizures in gliomas are more controversial. This protocol describes a systematic review and meta-analysis to clarify the prevalence and risk factors of early postoperative seizures in patients with glioma.

### Methods and analysis

Searches will be conducted on CNKI, WanFang, VIP, PubMed, Embase, Cochrane Library databases and Web of Science for the period from database inception to December 31st, 2023. Case-control and cohort studies of the incidence and risk factors for early postoperative seizures in all gliomas will be included. The primary outcome will be incidence, risk factors. Newcastle-Ottawa Scale was used for quality evaluation. Review of article screening, extracting data and risk of bias assessment will be repeated by two independent reviewers.

### Result

This study will provide evidence for the risk factors and incidence of early postoperative seizures in patients with glioma.

### Conclusion

Our study will provide evidence for the prevention of early postoperative seizures in glioma patients.

### Trail registration

This protocol was registered in PROSPERO and registration number is CRD42023415658.

**Funding:** The authors received no specific funding for this work.

**Competing interests:** I have read the journal's policy and the authors of this manuscript have the following competing interests: [The authors have declared that no competing interests exist.]

## Introduction

Glioma is a group of glial cell tumors of neuroepithelial origin, accounting for about 46% of all intracranial tumors [1, 2], which has a strong aggressiveness, high recurrence rate and mortality rate, and is one of the primary intracranial tumors with a poor prognosis commonly recognized in clinical practice [3, 4]. The World Health Organization (WHO) classification of central nervous system tumors classifies gliomas into low grade I and II gliomas and high grade III and IV gliomas, which mainly include astrocytoma, oligodendrogliomas, glioblastomas, and tumors of ventricular meningeal origin [5]. Most gliomas, except for grade I gliomas, have aggressive growth, poorly defined boundaries with the surrounding area, easy recurrence, and tend to destroy the neural network and structure of the surrounding tissues [6]. Early postoperative seizures is a group of syndromes caused by highly synchronized, often self-limiting abnormal discharges of neurons in the brain of known or unknown etiology [7]. Early postoperative seizures has been the most common and dominant clinical manifestation of glioma, mostly focal, but focal secondary generalized early postoperative seizures is not uncommon [8]. The risk of seizures secondary to glioma is high, approximately 75% for low-grade gliomas and 29%-49% for glioblastomas [9]. The mechanism by which early postoperative seizures occurs in gliomas, the most common intracranial tumor in clinical practice, remains unclear, but is simply due to an imbalance between inhibitory and agonistic signals within the cortex, and often the tumor itself is not the epileptic focus, while the surrounding normal cortical areas are the epileptic focus. Risk factors associated with secondary glioma attacks include age, gender, glioma type, WHO pathologic classification, glioma site, extent of resection, glioma size, glioma-cortical relationship, and presence of cystic calcification [10]. For example, high-grade gliomas. the risk of seizures was significantly lower in WHO grade III and IV glioblastomas than in low-grade gliomas, WHO grade I and II gliomas; gliomas located in the frontal and temporal lobes were more likely to cause early postoperative seizures [11], whereas gliomas located in the subepithelial and saddle areas were more likely to cause early postoperative seizures [12]. Gliomas located in the frontal and temporal lobes are more likely to cause early postoperative seizures, whereas gliomas located in the hypothalamus and saddle areas are less likely to cause early postoperative seizures [13, 14], and the closer the tumor is to functional areas and masses, the more likely it is to cause early postoperative seizures [15]. Isocitrate dehydrogenase 1 (IDH1) is also associated with seizure risk, and patients with IDH1 mutations in low-grade gliomas have a perioperative seizure risk high [16]. And there are some risk factors that are more controversial, such as the effect of multiple versus single lesions on seizures remains unclear [17, 18]. We present the first systematic review and meta-analysis protocol based on case-control and cohort studies to clarify the prevalence and risk factors of early postoperative seizures in patients with glioma.

## Material and methods

### Protocol and registration

The study is expected to start on 1 March 2023 and end on 31 December 2023. The review will be performed in compliance with the PRISMA guidelines (S1 Checklist) [19]. This protocol is registered with PROSPERO under registration number CRD42023415658 (S1 File).

### Literature selection

Inclusion criteria: diagnosed with glioma adults [20], The exposure factor was associated with early postoperative seizures, and the control group with no early postoperative seizures, The primary outcome will be incidence, risk factors, preoperative electroencephalogram (eeg) analysis; postoperative eeg analysis; this study will only consider Case-control and cohort studies.

The following studies will be excluded: duplicate articles, meta-analyses, reviews, protocols, animal studies, letters, no full text.

## Search strategy

The following electronic databases: PubMed, Web of Science, Cochrane Library, Embase will be searched from inception to April 20, 2023. The search strategy is shown in S2 File. The following search terms: early postoperative seizures, glioma, risk factors.

## Data collection and analysis

**Studies collection.** Two authors are requested to screen the retrieved researches independently. The repetitive researches will be removed, as well as the researches that do not meet the inclusion standards through reading titles and abstracts. The full text of each researches will be read with a view to selecting those that meet the inclusion standards. Any disagreements will be dealt with by means of discussing using a third reviewer. The total selection course of research selection is depicted inside Fig 1.

**Data extraction.** In the included researches, data extraction will be implemented independently by two reviewers according to one data acquisition list. That list will contain the foundational information (writer, journal, title, year of publication, alongside place of publication), research design, (research size), as well as conflicts of interest. If needful, one third reviewer will double-check the data with the intention of guaranteeing the consistence.

On conditions that data is incomplete or lost in any research, the writers will be contacted to attain the aforementioned data. With no available data, the research will be deleted.

**Assessment of risk of bias.** The bias risks of the included researches was evaluated independently by 2 investigators and the results were cross-checked. Evaluation of the bias risk for cohort and case-control researches was conducted via the Newcastle Ottawa scale (NOS) [21] to assess the quality of the literature. It contains 3 dimensions: selection of population, comparability, and exposure or outcome. This evaluation scale has 8 entries with a total of 9 points, with 0 to 4 low quality, 5 to 6 medium quality, as well as 7 to 9 elevated quality. Low quality literature with a score of 0 to 4 was not included in this study.

**Measures of treatment effect.** The Rvman5.4 software was adopted for statistical dissections. Besides, the stata15.0 software was adopted for the purpose of assessing the sensitivity and publication bias. As to consecutive consequence data, the Standard mean difference (SMD) or weighted mean difference (WMD) with 95% CI (Confidence Interval) will be adopted. As to dichotomous data, the risk ratio (RR) or odds ratio (OR) with 95% CI will be adopted for dissection.

**Assessment of heterogeneity.** Heterogeneity will be assessed via Cochrane 's Q test and the I2 statistic ($p > 0.05$ for Q statistic and I2 $< 50\%$ for statistical homogeneity). Since the researches might contain disparate follow-up times, meta-analyses will be implemented by means of one random effect model. Subgroup analyses will be conducted with the aim of detecting the potential reasons. On condition that heterogeneity is $> 75\%$, meta-analysis will not be conducted. One narrative, qualitative sum-up will be offered.

**Assess the quality of the evidence.** The accepted grading of recommendations assessment development and evaluation (GRADE) all over the world [22] was adopted for the sake of grading the quality of evidence for consequences. The overall RCTs were contained in this research. Those RCTs were designated as the most elevated degree of evidence. Additionally, there existed 5 elements which could lessen the evidence quality. The evidence quality: research constraints, inconsistent discoveries, indirectness of discoveries, imprecision of discoveries, bias of publication.

**PRISMA 2020 flow diagram for new systematic reviews which included searches of databases and registers only**

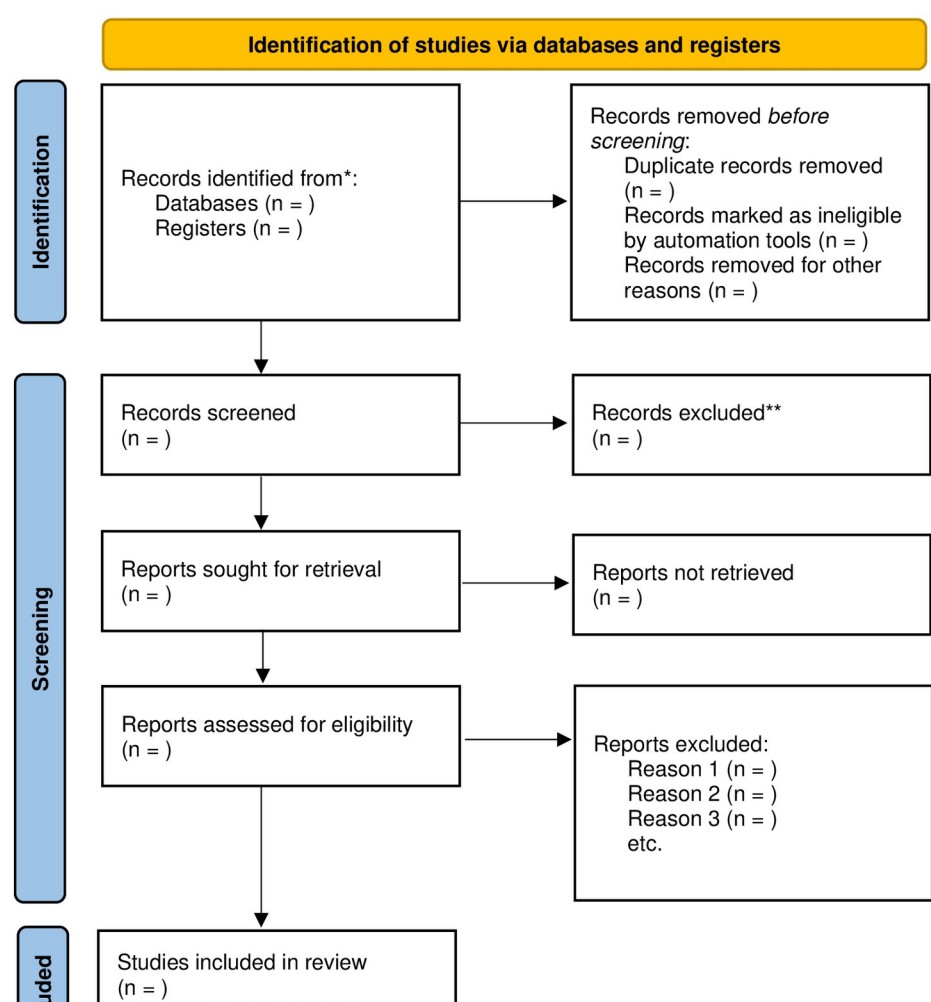

*From:* Page MJ, McKenzie JE, Bossuyt PM, Boutron I, Hoffmann TC, Mulrow CD, et al. The PRISMA 2020 statement: an updated guideline for reporting systematic reviews. BMJ 2021;372:n71. doi: 10.1136/bmj.n71

For more information, visit: http://www.prisma-statement.org/

**Fig 1. Flow diagram of the study design.** *Consider, if feasible to do so, reporting the number of records identified from each database or register searched (rather than the total number across all databases/registers). **If automation tools were used, indicate how many records were excluded by a human and how many were excluded by automation tools [36].

**Assessment of publication biases.** The inclusion of more than 10 studies was tested for publication bias via Egger's experiment and funnel plots. If there was asymmetry on both sides of the funnel plot, this indicated a higher likelihood of Egger's experiment and publication bias could be employed. On condition that the P value < 0.05, it indicates the existence of publication bias. Additionally, on condition that P > 0.05, it is the reverse funnel

plot consequence, at which point we are capable of further determining further using the cut-and-patch approach.

**Subgroup analysis.** On condition that data is available, subgroup dissection will be conducted with the objective to evaluating the heterogeneity in the light of the year of publication, country, type of study, age, gender.

## Sensitivity analysis

If possible, sensitivity dissection will be employed with the intention of assessing how unsure hypotheses of data and utilization impact the robustness of combined consequences. The specific influence of a paper upon the consequences of statistical dissection through eliminating literature one by one will be judged.

## Ethics and dissemination

This review will not request ethical approval since it does not encroach on anybody's interests. The consequences will be released in one peer-reviewed journal or propagated via conferences.

## Discussion

This systematic review will build on previous case-control and cohort studies of the incidence and risk factors for early postoperative seizures in all gliomas, and the conclusions drawn from this systematic review will be beneficial to patients and clinicians with glioma.

Most current studies suggest that a history of preoperative early postoperative seizures is one of the most important risk factors for early postoperative early postoperative seizures. One study found that patients with supratentorial brain tumors with a preoperative history of early postoperative seizures had a higher risk of early postoperative early postoperative seizures than those without a preoperative history of early postoperative seizures [23]. In a study by Bech et al [24], a history of preoperative seizures in patients with glioma increased the risk of postoperative seizures, and although surgical resection significantly reduced the incidence of seizures, a history of preoperative seizures and intraoperative seizures increased the risk of postoperative seizures. Although surgical resection significantly reduces the incidence of seizures, 29.2% and 80% of patients with a history of seizures and intraoperative seizures, respectively, develop seizures postoperatively [25]. Dewan et al [26] showed that patients without a history of preoperative early postoperative seizures were still prone to seizures after craniotomy, with an incidence rate of 7% to 18%. Therefore, a history of seizures before brain tumor surgery is an important risk factor for early seizure development after craniotomy [27]. The pathogenesis of mesenchymal tumor with early postoperative seizures is very complex and has not been fully elucidated. Recent studies indicate that mechanical compression of the tumor, hand-related injury, morphological changes of peritumoral tissue, peritumoral microenvironmental factors, and genetic changes in the tumor are closely related to the occurrence of early postoperative early postoperative seizures [28]. The high incidence of early postoperative seizures in low-grade glioma, meningioma and other low-grade brain tumors may be mainly related to the long course of such tumors, slow growth, large accumulation of tumors, and the formation of epileptogenic lesions caused by mechanical pressure (resulting in ischemia and hypoxia and other changes) repeatedly stimulating the normal peritumor brain tissue [29]. Study [30] have reported that 15% ~ 20% of epileptic seizures after surgical resection of cerebral edema are often associated with surgical trauma, acute anemic/hemorrhagic injury, cerebral edema. Daniel et al [31] found that long-term exposure to damaged cerebral cortex, venous sinus vessels, cortical vessels and cerebral cortex during surgery would significantly increase the risk of early postoperative early postoperative seizures.

Glioma is the most common primary intracranial central nervous system tumor in clinical practice [32]. Due to its increasing incidence, high lethality and poor prognosis, the disease burden of glioma in China is on the rise. Seizures are the main clinical manifestation or even the first symptom of glioma patients, which is related to the compression caused by the aggressive growth of tumor tissue and the abnormal discharge caused by the destruction of brain tissue; at the same time, early postoperative seizures is also a common complication in the early stage of postoperative patients with glioma, which is clinically manifested as general or localized limb convulsions, muscle twitching, incontinence, and sudden loss of consciousness [33, 34]. It causes great harm to the physical and mental health of patients It can seriously affect the quality of life of patients and even endanger their lives [35]. It is important to analyze the risk factors of early postoperative seizures after glioma surgery and take preventive measures according to the risk factors to improve patients' prognosis and provide quality of life.

The protocol has several advantages. We plan to search multiple databases using subject terms in conjunction with free words to ensure a comprehensive search of the literature. Another advantage is that strict inclusion and exclusion criteria will be used to ensure the quality of the included studies. In addition, we used GRADE ratings for quality assessment of outcome indicators.

## Supporting information

**S1 Checklist. PRISMA-P (Preferred Reporting Items for Systematic review and Meta-Analysis Protocols) 2015 checklist: Recommended items to address in a systematic review protocol\*.**
(DOC)

**S1 File. PROSPERO registration document.**
(PDF)

**S2 File. Retrieval strategy.**
(DOCX)

## Author Contributions

**Conceptualization:** Bo Sun, Wenpeng Lu, Peng Wang.

**Data curation:** Bo Sun, Wenpeng Lu, Ye Tian, Peng Wang.

**Formal analysis:** Bo Sun, Wangyang Yu, Ye Tian, Peng Wang.

**Funding acquisition:** Bo Sun, Ye Tian, Peng Wang.

**Investigation:** Bo Sun, Wenpeng Lu, Wangyang Yu, Peng Wang.

**Methodology:** Bo Sun, Wenpeng Lu, Wangyang Yu, Peng Wang.

**Project administration:** Bo Sun, Wenpeng Lu, Wangyang Yu, Ye Tian, Peng Wang.

**Resources:** Ye Tian, Peng Wang.

**Software:** Bo Sun.

**Supervision:** Wangyang Yu.

**Validation:** Bo Sun, Wenpeng Lu, Wangyang Yu.

**Visualization:** Wenpeng Lu, Wangyang Yu.

**Writing – original draft:** Bo Sun, Wenpeng Lu.

**Writing – review & editing:** Bo Sun, Wenpeng Lu, Wangyang Yu.

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
