## [Decision Letter · Decision Letter 0]

18 Dec 2023

PONE-D-23-24158Prevalence and risk factors of epilepsy in glioma: A protocol for meta-analysis and systematic reviewPLOS ONE

Dear Dr. wang,

Thank you for submitting your manuscript to PLOS ONE. After careful consideration, we feel that it has merit but does not fully meet PLOS ONE’s publication criteria as it currently stands. Therefore, we invite you to submit a revised version of the manuscript that addresses the points raised during the review process.

We look forward to receiving your revised manuscript.

Kind regards,

Jianhong Zhou

Staff Editor

PLOS ONE

 [None]. 

4. PLOS requires an ORCID iD for the corresponding author in Editorial Manager on papers submitted after December 6th, 2016. Please ensure that you have an ORCID iD and that it is validated in Editorial Manager. To do this, go to ‘Update my Information’ (in the upper left-hand corner of the main menu), and click on the Fetch/Validate link next to the ORCID field. This will take you to the ORCID site and allow you to create a new iD or authenticate a pre-existing iD in Editorial Manager. Please see the following video for instructions on linking an ORCID iD to your Editorial Manager account: " ext-link-type="uri" xlink:type="simple">https://www.youtube.com/watch?v=_xcclfuvtxQ".

Additional Staff Editor Comments: PLOS ONE recently began considering Study Protocols, an article type where proposed work is reviewed prior to completion. Two of the reviewers have concerns about the article type. Their comments are acknowledged. But this study is within scope of PLOS ONE. Please see our guidelines for Study Protocols (https://journals.plos.org/plosone/s/submission-guidelines#loc-study-protocols).

While the reviewers acknowledged the efforts of the authors, they were largely of the opinion that the protocol is not well suited for PLOS One. The authors are advised to strengthen the manuscript with the reviewers' comments and submit to an alternate journal.

Reviewers' comments:

Reviewer's Responses to Questions

**Comments to the Author**

1. Does the manuscript provide a valid rationale for the proposed study, with clearly identified and justified research questions?

Reviewer #1: No

Reviewer #2: Yes

Reviewer #3: Yes

2. Is the protocol technically sound and planned in a manner that will lead to a meaningful outcome and allow testing the stated hypotheses?

Reviewer #1: Yes

Reviewer #2: Yes

Reviewer #3: Yes

3. Is the methodology feasible and described in sufficient detail to allow the work to be replicable?

Reviewer #1: Yes

Reviewer #2: Yes

Reviewer #3: Yes

4. Have the authors described where all data underlying the findings will be made available when the study is complete?

Reviewer #1: No

Reviewer #2: Yes

Reviewer #3: Yes

5. Is the manuscript presented in an intelligible fashion and written in standard English?

Reviewer #1: Yes

Reviewer #2: Yes

Reviewer #3: Yes

6. Review Comments to the Author

You may also provide optional suggestions and comments to authors that they might find helpful in planning their study.

Reviewer #1: Please specify the timing of the seizure onset: Was it preoperative, postoperative, or during the follow-up?

I believe a protocol solely for meta-analysis might not be suitable for publication in PLOS ONE, given its reputation and impact factor.

Reviewer #2: The relevant part of the article is the macroscopic structural analysis of gliomas in their different degrees and their relationship with epilepsy. Which would allow us to predict the risk of an event. As a recommendation, document what is reported in the literature according to the pre- and postoperative electroencephalographic analysis. And I would like to know if there is any relationship between epilepsy and tumor behavior in spectroscopy and tractography.

Reviewer #3: The protocol seems reasonable. But I will need to see a manuscript before i can evaluate if the manuscript is suitable for publication.

7. PLOS authors have the option to publish the peer review history of their article (what does this mean?). If published, this will include your full peer review and any attached files.

Reviewer #1: No

Reviewer #2: **Yes: **Manuel Eduardo Soto Garcia

Reviewer #3: No

---

## [Author Response · Author response to Decision Letter 0]

20 Dec 2023

Dear editor and dear reviewers

Re Manuscript ID (PONE-D-23-24158) entitled " Prevalence and risk factors of epilepsy in glioma: A protocol for meta-analysis and systematic review". Thank you for your letter and for the reviewer’s comments. Those comments are all valuable and very helpful for revising and improving our paper, as well as the important guiding significant to our research. We have studies comments carefully and have made correction which we hope to meet with approval. Revised portions are marked in red in the paper. The main correction in the paper and responds to the reviewer’s are as flowing:

Reviewer #1: Please specify the timing of the seizure onset: Was it preoperative, postoperative, or during the follow-up? I believe a protocol solely for meta-analysis might not be suitable for publication in PLOS ONE, given its reputation and impact factor. 

responds: Thank you very much for the valuable comments of the reviewer, and we made it clear that it was the early postoperative seizure. However, according to our search “((protocol[Title]) AND (meta-analysis[Title/Abstract])) AND ("PloS one"[Journal])” . There are still 86 articles on meta-analysis protocol from January to December 2023, so I do not agree with the last part of reviewer 1's comments.

Reviewer #2: The relevant part of the article is the macroscopic structural analysis of gliomas in their different degrees and their relationship with epilepsy. Which would allow us to predict the risk of an event. As a recommendation, document what is reported in the literature according to the pre- and postoperative electroencephalographic analysis. And I would like to know if there is any relationship between epilepsy and tumor behavior in spectroscopy and tractography.

responds: Epilepsy and brain tumors in patients with preoperative and postoperative electroencephalogram (EEG) analysis has been a considerable research subject. Here is a summary of the findings:

Preoperative electroencephalogram (eeg) analysis:

Location of epileptic foci: eeg is helpful to determine the specific brain region epileptic seizures.

Characterizations of seizure activity: the EEG records can distinguish between different types of attacks (e.g., focal or comprehensiveness), and help to understand the patterns.

Surgical planning: EEG data help neurosurgeons plan surgery, guiding them to the precise resection area to minimize damage to key brain regions.

Postoperative eeg analysis:

Evaluation of surgical efficacy: Postoperative EEG helps to assess the success of surgical interventions in reducing or eliminating epileptic activity.

Residual activity monitoring: Used to monitor residual abnormal electrical activity after surgery.

On the relationship between epilepsy and tumor behavior in spectroscopy and tractography:

Spectral analysis:

Metabolic changes: Spectroscopy, such as magnetic resonance spectroscopy (MRS), can identify metabolic changes in the brain associated with epilepsy and tumors. For example, it can detect increased levels of certain metabolites such as lactate or decreased levels of N-acetylaspartate (NAA).

Distinguishing tissue features: By analyzing metabolite concentrations, it is helpful to distinguish tumor tissues from epileptogenic tissues.

.

Reviewer #3: The protocol seems reasonable. But I will need to see a manuscript before i can evaluate if the manuscript is suitable for publication.

responds: Thank you very much for referees, enclosed is our the body of the manuscript.

Prevalence and risk factors of early postoperative seizures in patients with glioma: a systematic review and meta-analysis

Running title: risk factors of seizures in glioma

Abstract

Objective: To explore the prevalence of and risk factors for early postoperative seizures in patients with glioma through a systematic review and meta-analysis.

Methods: Case-control studies on the prevalence and risk factors of early postoperative seizures in patients with glioma were retrieved from CNKI, Wan Fang, VIP, PubMed, Embase, Cochrane Library and Web of Science databases, and the retrieval deadline was April 1st, 2023. Stata15.0 was used for the data analysis.

Results: Eleven studies were case-control studies including 488 patients with early postoperative seizures and 2,051 patients without early postoperative seizures. These results suggest that the prevalence of glioma complicated by seizures (ES = 19%, 95% confidence interval [CI] [14%–25%]). The results of the meta-analysis showed: history of seizures, dyskinesia, frontal lobe tumor, pathological grade ≤2, tumor≥ 3 cm, tumor resection, tumor edema ≥ 2 cm, and glioma cavity hemorrhage. The multivariate analysis results showed: history of seizures, dyskinesia, tumor ≥3 cm, peritumoral edema ≥2 cm and glioma cavity hemorrhage were risk factors for glioma complicated with early postoperative seizures.

Conclusion: Under the existing evidence, seizures history, dyskinesia, frontal lobe tumor, pathological grade ≤2, tumor ≥3 cm, partial tumor resection, edema around tumor ≥2 cm, and glioma cavity hemorrhage are risk factors for glioma complicated with early postoperative seizures.

Key words: glioma, seizures, risk factors, meta-analysis, systematic review

1. Background

Gliomas originate from the abnormal proliferation of glial cells in the brain (1, 2) and are common primary central nervous system tumors, accounting for 50% to 60% of primary intracranial tumors (3), with a high mortality rate (4). The annual incidence of glioma is approximately 5–8 cases per 100,000 people, and the annual mortality rate is as high as 30,000, ranking third after pancreatic and lung cancer (5-7). In patients with gliomas, seizures are often the initial symptom and the reason for seeking medical attention. Glioma-related seizures is one of the most common accompanying symptoms in patients with glioma, especially in low-grade gliomas (8, 9), where the main symptoms are sudden paroxysmal convulsions, foaming at the mouth, loss of consciousness, and convulsions of the limbs (10). The specific mechanisms underlying glioma-related seizures are not yet fully understood, and pathological changes and pathogenic mechanisms are diverse, likely resulting from the combined effects of multiple factors (11, 12). Abnormal expression of tumor related genes is believed to be closely associated with the occurrence and development of seizures in brain tumors, especially gliomas (13). Although advances in surgery, radiotherapy, and chemotherapy have improved the prognosis of patients with glioma, the presence of seizures severely affects their quality of life (14). Currently, the clinical efficacy of treating glioma-related seizures remains unsatisfactory because of an incomplete understanding of the pathophysiological, biochemical, molecular, and pharmacological mechanisms underlying its occurrence (15, 16). Giraldi et al. (17) reported a significantly increased cumulative risk of new-onset seizures after craniotomy. Previous studies confirmed that seizures is a common complication of intracranial gliomas that can lead to disability and even death. Therefore, it is important to explore the relevant risk factors for early postoperative seizures in patients with gliomas. However, risk factors for early postoperative seizures in patients with glioma remain controversial (18). Therefore, this study aimed to resolve these controversies by conducting a meta-analysis and systematic review to investigate risk factors for seizures in patients with gliomas. It is hoped that by preventing and treating early postoperative seizures, the quality of life and prognosis of patients with glioma can be improved.

2. Materials and methods

This scheme is based on the preferred reporting items of the Protocol for Systematic Reviews and Meta-analyses (PRISMA-P). The review will be conducted according to PRISMA criteria (19). The registration number was CRD42023415658.

2.1 Literature search

We searched the CNKI, Wan Fang, VIP, PubMed, Embase, Cochrane Library, Web of Science, and other databases on the incidence and risk factors of seizures in patients with glioma. The search deadline was April 1, 2023. Subjects and free words were used for retrieval: seizures, glioma, and risk factors. See Supplementary Material 1 for the specific retrieval strategies.

2.2 Inclusion and exclusion criteria

The inclusion criteria were adults who met the diagnostic criteria for glioma (including low- and high-grade gliomas) (20), and the exposure factor was a case-control study of early postoperative seizures (defined as seizures occurring within one week after surgery). The primary outcome measures were univariate and multivariate risk factor analyses. The analysis can include both univariate and multivariate results. However, when both univariate and multivariate results are reported in the same study, multivariate results should be prioritized. The adjusted risk ratio from the multivariate analysis was extracted. If only univariate analysis is reported, the univariate results should be extracted. The secondary outcome measure was the prevalence of seizures in patients with gliomas.

The exclusion criteria were conference abstracts, meta-analyses, protocols, letters, repeatedly published articles, systematic reviews, failure to obtain full text, failure to obtain available data, and animal experiments.

2.3 Data extraction

Two independent evaluators independently screened the literature to extract the data and directly screened the easily judged literature by reading the titles and abstracts of the literature and the full text; for any disagreements, they consulted relevant experts. The inclusion and exclusion criteria were strictly followed during the screening process. They extracted the corresponding indicators from the studies and crosschecked the extracted data to ensure consistency. The main data extracted included the name of the first author, year of publication, country, study design, sample size, sex, and age.

2.4 Quality evaluation

The Newcastle-Ottawa Scale (NOS) (21) was used to evaluate case-control studies, including the selection of the study population (4 points), comparability between groups (2 points), and measurement of exposure factors or results (3 points). The total score of the scale is 9, with ≤4 indicating low quality, 5–6 indicating medium quality, and ≥7 indicating high quality. If the two researchers disagree on the evaluation process, they will discuss the decision or ask a third party to decide.

2.5 Statistical analysis

Stata 15.0 was used to statistically analyze the data. The risk values for each study were described using the RR values and 95% confidence intervals (CIs) were calculated. The heterogeneity test (Q test) and I2 statistics were used to select the appropriate model for calculating the pooled RR. If I2 was greater than 50%, the random effects model was adopted; if I2 was less than or equal to 50%, the fixed effects model was adopted. For I2 50%, we assessed the sensitivity of the literature using the leave-one-out method. Additionally, we conducted a publication bias using the Egger test, with significance level set at α = 0.05. A P-value 0.05 was considered statistically significant.

3. Results

3.1 Results of the literature retrieval

By searching the CNKI, Wan Fang, VIP, PubMed, Embase, Cochrane Library, and Web of Science databases, 641 documents were initially obtained, 503 documents were obtained by removing duplicate documents, 36 articles were preliminarily screened by reading titles and abstracts, and 12 documents were finally included after reading the full text. Figure 1 shows the retrieval flowchart.

3.2 Basic features of the included literature

The 11 included (22-32) studies were case-control studies, including 488 patients with early postoperative seizures and 2,051 patients without early postoperative seizures; the age of inclusion ranged from 7 to 81 years. See Supplementary Material 2 and Table S1 for specific document characteristics. Twelve articles were evaluated by NOS quality: one (23) scored six points, and the research quality was moderate. The rest scored 7–8 points, and the overall quality of the included studies was high. See Supplementary Material 2 and Table S2 for the specific quality evaluations.

3.3 Prevalence of glioma with seizures

Eleven studies have reported the prevalence of gliomas complicated by seizures. The heterogeneity test (I2 = 92.3%, P = 0.001) was analyzed using a random-effects model, and the results suggested that the prevalence of glioma complicated with seizures (ES = 19%, 95% CI [14%–25%]) because the heterogeneity of the indicators was large; therefore, sensitivity analysis was carried out to eliminate them one by one. The analysis results indicated that the sensitivity was low, and the analysis results were stable. The Egger test was performed on the index to evaluate publication bias (P = 0.026) and the prevalence was more likely to indicate publication bias (see Supplementary Material 3).

3.4 Single factor meta-analysis

3.4.1 Seizures history

Nine studies mentioned a history of seizures (defined as tumor-related seizures before surgery) as a risk factor for seizures, and the heterogeneity test (I2 = 24.6%, P = 0.225) was analyzed by the fixed-effect model. The results suggested that a history of seizures was a risk factor for glioma complicated by seizures, and the difference was statistically significant (RR = 1.94, 95% CI [1.76,2.14], P = 0.001). See Figure 2 and Supplementary Material 2 Table S3.

3.4.2 Dyskinesia

Dyskinesia was mentioned as a risk factor for seizures in three studies, and the heterogeneity test (I2 = 89.2%, P = 0.001) was analyzed using a random-effects model. The results suggested that dyskinesia was a risk factor for glioma complicated by seizures, and the difference was statistically significant (RR = 3.13,95% CI [1.20,8.15], P = 0.02). See Figure 3 and Supplementary Material 2 Table S3.

3.4.3 Frontal tumor

Nine studies identified frontal lobe tumor location as a risk factor for seizures, and the heterogeneity test (I2 = 75.6%, P = 0.001) was analyzed using a random-effects model. The results showed that frontal lobe tumors were risk factors for glioma complicated with seizures, and the difference was statistically significant (RR = 1.45, 95% CI [1.16,1.83], P = 0.001), as shown in Figure 4 and Supplementary Material 2 Table S3.

3.4.4 Pathological grade ≤2

Eight studies referred to a pathological grade ≤2 as a risk factor for seizures, and the heterogeneity test (I2 = 85.9%, P = 0.001) was analyzed using a random-effects model. The results of the analysis suggested that pathological grade ≤2 was a risk factor for tumor-associated seizures and the difference was statistically significant (RR = 1.74, 95% CI [1.13, 2.67], P = 0.012). See Figure 5 and Supplementary Material 2 Table S3.

3.4.5 Tumor ≥3 cm 

Eight studies mentioned tumors ≥3 cm as a risk factor for seizures and the heterogeneity test (I2 = 92.4%, P = 0.001) was analyzed using a random-effects model, which suggested that tumors ≥3 cm were a statistically significant difference as a risk factor for tumor-associated seizures (RR = 1.70, 95% CI [1.18, 2.45], P = 0.005). See Figure 6 and Supplementary Material 2 Table S3.

3.4.6 Partial resection of tumor

Seven studies mentioned partial tumor resection as a risk factor for seizures, and the heterogeneity test (I2 = 7.7%, P = 0.369) was analyzed using a fixed-effects model. The results suggested that partial tumor resection was a statistically significant risk factor for tumor-associated seizures (RR = 1.60, 95% CI [1.36,1.88], P = 0.001). See Supplementary Material 2 Table S3).

3.4.7 Peritumor edema ≥2 cm

Six studies mentioned peritumor edema ≥2 cm as a risk factor for seizures and the heterogeneity test (I2 = 63.7%, P = 0.017) was analyzed using a random-effects model, the results of which suggested that peritumor edema ≥2 cm was a statistically significant difference as a risk factor for tumor-associated seizures (RR = 1.77,95% CI [1.40, 2.25], P = 0.001). See Supplementary Material 2 Table S3.

3.4.8 Intracavitary hemorrhage of glioma

Six studies mentioned glioma cavity hemorrhage as a risk factor for seizures, and the heterogeneity test (I2 = 83.9%, P = 0.001) was analyzed using a random-effects model; the results suggested that glioma cavity hemorrhage was a risk factor for tumor-associated seizures, and the difference was statistically significant (RR = 3.15, 95% CI (1.85, 5.37), P = 0.001. See Supplementary Material 2 Table S3.

3.4.9 Other meta-analysis results

Differences in the correlations between male and female sex, circulatory disease, metabolic disease, blurred tumor boundaries, prophylactic medication, and glioma with seizures were not statistically significant (Supplementary Material 2, Table S3).

3.4.10 Multi-factor meta results

The results of the multifactorial analysis mentioned in the research study were analyzed and combined, and the results suggested that there was no statistically significant difference in the correlation between frontal lobe tumors, partial tumor resection, prophylactic medication, and tumor-associated seizures. History of seizures (ES = 2.54, 95% CI [1.24, 5.20], P = 0.011), motor impairment (ES = 2.53, 95% CI (1.83, 3.51), P = 0.001], tumor ≥3 cm (ES = 2.56, 95% CI [1.99, 3.31], P = 0.001), peritumor edema ≥2 cm (ES = 2.53, 95% CI [1.83, 3.51], P = 0.001), and glioma cavity hemorrhage (ES = 2.93, 95% CI (1.79, 4.81), P = 0.001) were risk factors for tumor-associated seizures (see Supplementary Material 2 Table S4).

4. Publication bias

The publication bias was evaluated using Egger's test for each risk factor and the P0.05 for each indicator, whether univariate or multifactorial, suggesting that there was no publication bias (see Supplementary Material 2 Tables S3 and S4).

5. Discussion

This study is the first to explore the risk factors for gliomas complicated by seizures using a meta-analysis. Through single and multiple factors, this study found that a history of seizures is an independent risk factor for glioma complicated by seizures, indicating that the incidence of seizures in patients with brain glioma with preoperative seizures increased significantly. This may be related to the instability of neuronal membrane potential, decrease in seizure threshold, and abnormal discharge in patients with seizures, which is caused by abnormal and excessive discharge of cortical neurons. Therefore, patients with preoperative seizures should actively use drugs to reduce its occurrence (36). The present study also found a significantly higher incidence of seizures in patients with dyskinesia in cranial glioma; However, previous literature found that there was no strong correlation between early postoperative epilepsy and dyskinesia(37), This conclusion may be due to the small number of dyskinesia included in our study, but this is also what we should pay attention to in future studies. The incidence of seizures is significantly higher in patients with peri-tumoral edema ≥2 cm. A possible reason for this is that peritumoral edema includes both vasogenic and cytotoxic brain edema. Vasogenic brain edema increases capillary permeability and disrupts the blood-brain barrier, leading to an abnormal distribution of ions inside and outside the cells, which in turn stimulates nerves and triggers seizures (38). In addition, when astrocytes are involved in cytotoxic brain edema due to increased occupancy effects, their ability to take up glutamate is reduced, which in turn affects the stability of the nerve cell membrane, inducing neuroexcitability, and ultimately seizures (39). This study also found that intratumoral hemorrhage is an independent risk factor for gliomas complicated by seizures. Intratumoral hemorrhage can cause brain tissue hypoxia and energy metabolism disorders. Iron ions in the blood can catalyze the production of oxygen free radicals, form lipid peroxides, and cause neuronal necrosis, leading to seizures. Therefore, intratumoral hemorrhage is considered a cause of early seizures (40). It has been shown that tumor pathological grade is usually negatively correlated with early postoperative seizure events, possibly because seizures are characterized by abnormal neuronal firing, and as the pathological grade of the glioma increases, infiltrative growth tends to become more pronounced, and this infiltrative growth of tumor cells may destroy projection fibers and neurons, thus inhibiting the spread of seizure firing (41). This is consistent with the findings of the present study that pathological grade ≤2 is an independent risk factor for tumor-associated seizures. In patients with larger tumors, the tumor is more likely to compress adjacent normal brain tissues, causing ischemia and metabolic disorders in the brain, imbalance of intra- and extracellular ion levels, and increasing the risk of seizures, which further supports the findings of the present study that tumors ≥3 cm are an independent risk factor for tumor-associated seizures (42, 43). Telfeian et al. observed that the smaller the tumor size of glioblastoma multiforme, the higher the risk of postoperative seizures. This could be because more brain tissue dissection may be required to reach smaller tumors, thus explaining the higher risk of seizures after surgery for smaller tumors (44). In patients with partially resected gliomas, the residual tumor may continue to invade brain tissue and stimulate the cerebral cortex, leading to abnormal cortical discharges and triggering seizures (45). Therefore, the patient's condition should be considered during clinical treatment, and the tumor should be removed as completely as possible; if this is not possible, early prevention and treatment of seizures should be actively pursued before and after surgery to reduce the risk ofearly postoperative seizures and improve patient prognosis.

The current study has the following limitations: first, the number of included articles is small and most of them are from China, which may be subject to selection bias; second, the diagnostic criteria for glioma and seizures used in the included studies were not consistent, which may account for the large heterogeneity; third, the analysis process of this study does not distinguish between OR, RR, or HR, and although the actual difference between the three is not significant, there is some difference in the risk of the three essentially measured diseases, which may lead to some bias in the results.

6. Conclusion

Based on the available evidence, history of seizures, dyskinesia, frontal lobe tumor, pathological grade ≤2, tumor ≥3 cm, partial tumor resection, peritumor edema ≥2 cm, and glioma cavity hemorrhage are risk factors for early postoperative seizures, and clinical practitioners can combine these indicators for early detection, diagnosis, and intervention in patients with early postoperative seizures, thereby improving the quality of life of such patients.

References

1. Yan Y, Zeng S, Gong Z, Xu Z. Clinical implication of cellular vaccine in glioma: current advances and future prospects. J Exp Clin Cancer Res. 2020;39(1):257.

2. Lang F, Liu Y, Chou FJ, Yang C. Genotoxic therapy and resistance mechanism in gliomas. Pharmacol Ther. 2021;228:107922.

3. Yang K, Wu Z, Zhang H, Zhang N, Wu W, Wang Z, et al. Glioma targeted therapy: insight into future of molecular approaches. Mol Cancer. 2022;21(1):39.

4. Norouzi M. Gold nanoparticles in glioma theranostics. Pharmacol Res. 2020;156:104753.

5. Lin J, Bytnar JA, Theeler BJ, McGlynn KA, Shriver CD, Zhu K. Survival among patients with glioma in the US military health system: a comparison with patients in the Surveillance, Epidemiology, and End Results program. Cancer. 2020;126(13):3053-60.

6. Roux A, Boddaert N, Grill J, Castel D, Zanello M, Zah-Bi G, et al. High prevalence of developmental venous anomaly in diffuse intrinsic pontine gliomas: a pediatric control study. Neurosurgery. 2020;86(4):517-23.

7. Bello-Alvarez C, Camacho-Arroyo I. Impact of sex in the prevalence and progression of glioblastomas: the role of gonadal steroid hormones. Biol Sex Differ. 2021;12(1):28.

8. Berg-Beckhoff G, Schüz J, Blettner M, Münster E, Schlaefer K, Wahrendorf J, et al. History of allergic disease and seizures and risk of glioma and meningioma (INTERPHONE study group, Germany). Eur J Epidemiol. 2009;24(8):433-40.

9. Douw L, de Groot M, van Dellen E, Aronica E, Heimans JJ, Klein M, et al. Local MEG networks: the missing link between protein expression and seizures in glioma patients? Neuroimage. 2013;75:195-203.

10. Armstrong TS, Grant R, Gilbert MR, Lee JW, Norden AD. Seizures in glioma patients: mechanisms, management, and impact of anticonvulsant therapy. Neuro Oncol. 2016;18(6):779-89.

11. Yamagata A, Fukai S. Insights into the mechanisms of seizures from structural biology of LGI1-ADAM22. Cell Mol Life Sci. 2020;77(2):267-74.

12. Schlehofer B, Blettner M, Moissonnier M, Deltour I, Giles GG, Armstrong B, et al. Association of allergic diseases and seizures with risk of glioma, meningioma and acoustic neuroma: results from the INTERPHONE international case-control study. Eur J Epidemiol. 2022;37(5):503-12.

13. Fang S, Li L, Weng S, Guo Y, Fan X, Jiang T, et al. Altering patterns of sensorimotor network in patients with different pathological diagnoses and glioma-related seizures under the latest glioma classification of the central nervous system. CNS Neurosci Ther. 2023;29(5):1368-78.

14. Avila EK, Chamberlain M, Schiff D, Reijneveld JC, Armstrong TS, Ruda R, et al. Seizure control as a new metric in assessing efficacy of tumor treatment in low-grade glioma trials. Neuro Oncol. 2017;19(1):12-21.

15. de Bruijn M, van Sonderen A, van Coevorden-Hameete MH, Bastiaansen AEM, Schreurs MWJ, Rouhl RPW, et al. Evaluation of seizure treatment in anti-LGI1, anti-NMDAR, and anti-GABA(B)R encephalitis. Neurology. 2019;92(19):e2185-e96.

16. van der Meer PB, Dirven L, Fiocco M, Vos MJ, Kouwenhoven MCM, van den Bent MJ, et al. First-line antiepileptic drug treatment in glioma patients with seizures: levetiracetam vs valproic acid. Epilepsia. 2021;62(5):1119-29.

17. Giraldi L, Vinsløv Hansen J, Wohlfahrt J, Fugleholm K, Melbye M, Munch TN. Postoperative de novo seizures after craniotomy: a nationwide register-based cohort study. J Neurol Neurosurg Psychiatry. 2022;93(4):436-44.

18. Koekkoek JA, Dirven L, Taphoorn MJ. The withdrawal of antiepileptic drugs in patients with low-grade and anaplastic glioma. Expert Rev Neurother. 2017;17(2):193-202.

19. Liberati A, Altman DG, Tetzlaff J, Mulrow C, Gøtzsche PC, Ioannidis JP, et al. The PRISMA statement for reporting systematic reviews and meta-analyses of studies that evaluate health care interventions: explanation and elaboration. PLoS Med. 2009;6(7):e1000100.

20. Jiang T, Nam DH, Ram Z, Poon WS, Wang J, Boldbaatar D, et al. Clinical practice guidelines for the management of adult diffuse gliomas. Cancer Lett. 2021;499:60-72.

21. Stang A. Critical evaluation of the Newcastle-Ottawa scale for the assessment of the quality of nonrandomized studies in meta-analyses. Eur J Epidemiol. 2010;25(9):603-5.

22. Chen Yingdong, Peng Biao, LUO Dongdong, SONG Qingyu. Risk factors for early epileptic seizures after brain glioma surgery. Chinese Journal of Microinvasive Neurosurgery. 2019; 24 (5) : 204-7

23. Deng Hui, HU Jiadan, LIU Jia, ZHU Wenjun, HU Jin. Risk factors forearly postoperative seizures in patients with glioma. Medical equipment. 2022; 35 (17) : 97-9..

24. GUI Xuebao, FU Xianming, NIU Chaosh, QIAN Ruobin, YAO Yang, ZHU Siyang, et al. Risk factors of early epileptic seizure after supratentorial glioma surgery. Journal of Stereotactic and Functional Neurosurgery. 2012; 25 (5) : 287-90.

25. Hasmujiang Reheman, Yang Xiaopeng. Analysis of the status quo and risk factors of seizures in early stage after brain glioma surgery. Cancer Progress. 2017; 15 (8) : 947-9.

26. Hu Juntao, LV Yan-Xia, WANG Hui, TU Hanjun, LI Xinjian, HU Shengli, et al. Analysis of influencing factors of seizures after brain glioma surgery. Shaanxi Medical Journal. 2014(4):471-3.

27. Tian Cong, WU Runqiu, ZHANG Xu, Yu Yas. Analysis of related influencing factors of epileptic seizure after craniocerebral glioma surgery. China Medical Journal. 2022; 57 (12) : 1337-40.

28. Yang Shaowei, XIA Lei, GAO Changqing, Xie Yu, WANG Anbang, DONG Developed. Analysis of risk factors for early epileptic seizure after brain glioma surgery. Chinese Journal of Applied Neurological Diseases. 2015(24):24-6.

29. Zhang P, Lu Y. Analysis of related factors of early epileptic seizure after brain glioma surgery. Chinese Journal of Applied Neurological Diseases. 2014(7):68-9,70.

30. Zhang Xiaocong, MA Junpeng, ZHU Chuangye, Yu Qiang, Chen Ken, Jiang Jie. Logistic regression analysis of risk factors associated with early epileptic seizure after brain glioma surgery. Cancer prevention and Treatment. 2021; 34 (02) : 138-42.

31. Yuen TI, Morokoff AP, Bjorksten A, D'Abaco G, Paradiso L, Finch S, et al. Glutamate is associated with a higher risk of seizures in patients with gliomas. Neurology. 2012;79(9):883-9.

32. Yu Z, Zhang N, Hameed NUF, Qiu T, Zhuang D, Lu J, et al. The analysis of risk factors and survival outcome for Chinese patients with seizures with high-grade glioma. World Neurosurgery. 2019;125:e947-e57.

33. Falco-Walter J. Seizures-definition, classification, pathophysiology, and epidemiology. Semin Neurol. 2020;40(6):617-23.

34. Specchio N, Wirrell EC, Scheffer IE, Nabbout R, Riney K, Samia P, et al. International League Against Seizures classification and definition of seizures syndromes with onset in childhood: Position paper by the ILAE Task Force on Nosology and Definitions. Epilepsia. 2022;63(6):1398-442.

35. Reddy C, Saini AG. Metabolic Seizures. Indian J Pediatr. 2021;88(10):1025-32.

36. Leone MA, Ivashynka AV, Tonini MC, Bogliun G, Montano V, Ravetti C, et al. Risk factors for a first epileptic seizure symptomatic of brain tumour or brain vascular malformation. A case control study. Swiss Med Wkly. 2011;141:w13155.

37. Fang S, Li Y, Wang Y, Zhang Z, Jiang T. Awake craniotomy for gliomas involving motor-related areas: classification and function recovery. J Neurooncol. 2020;148(2):317-25.

38. Gao A, Yang H, Wang Y, Zhao G, Wang C, Wang H, et al. Radiomics for the prediction of seizures in patients with frontal glioma. Front Oncol. 2021;11:725926.

39. Mader MM, Deuter D, Sauvigny T, Borchert P, Faizy TD, Bester M, et al. Diffusion tensor imaging changes in patients with glioma-associated seizures. J Neurooncol. 2022;160(2):311-20.

40. Zhang R, Xu X, Zhou H, Yao D, Wei R, Muhammad S. Pediatric angiocentric glioma with acute intracerebral hemorrhage: a case report with 36 months follow-up. Surg Neurol Int. 2021;12:499.

41. Luyken C, Blümcke I, Fimmers R, Urbach H, Elger CE, Wiestler OD, et al. The spectrum of long-term seizures-associated tumors: long-term seizure and tumor outcome and neurosurgical aspects. Epilepsia. 2003;44(6):822-30.

42. Slegers RJ, Blumcke I. Low-grade developmental and seizures associated brain tumors: a critical update 2020. Acta Neuropathol Commun. 2020;8(1):27.

43. Métais A, Appay R, Pagès M, Gallardo C, Silva K, Siegfried A, et al. Low-grade seizures-associated neuroepithelial tumours with a prominent oligodendroglioma-like component: the diagnostic challenges. Neuropathol Appl Neurobiol. 2022;48(2):e12769.

44. Telfeian AE, Philips MF, Crino PB, Judy KD.early postoperative seizures in patients undergoing craniotomy for glioblastoma multiforme. J Exp Clin Cancer Res. 2001;20(1):5-10.

45. Grajkowska W, Matyja E, Daszkiewicz P, Roszkowski M, Peregud-Pogorzelski J, Jurkiewicz E. Angiocentric glioma: a rare intractable seizures-related tumour in children. Folia Neuropathol. 2014;52(3):253-9.

Figure 1 Flow chart of the literature search

Figure 2 Forest plot of the meta-analysis of seizures history 

Figure 3 Forest plot of the meta-analysis of dyskinesia 

Figure 4 Forest plot of the meta-analysis of frontal lobe tumors

Figure 5 Forest plot of the meta-analysis of pathological grade 2 

Figure 6 Forest plot of the meta-analysis of tumor 3 cm 

Supplementary Material 1 Search history

Supplementary Material 2 (Table S1 Literature characteristics; Table S2 NOS scores; Table S3 Single-factor meta-analysis; Table S4 multi-factor meta-analysis).

Supplementary Material 3 (A: Forest plot for incidence meta-analysis; B: Incidence sensitivity analysis; C: Incidence Egger’s test).

---

## [Editor Report · Decision Letter 1]

17 Mar 2024

Prevalence and risk factors of early postoperative seizures in patients with glioma: A protocol for meta-analysis and systematic review

PONE-D-23-24158R1

Dear Dr. wang,

We’re pleased to inform you that your manuscript has been judged scientifically suitable for publication and will be formally accepted for publication once it meets all outstanding technical requirements.

Kind regards,

Aurel Popa-Wagner

Academic Editor

PLOS ONE

Additional Editor Comments (optional):

The authors have adequately addressed the Reviewers' comments.
---

## [Editor Report · Acceptance letter]

22 Mar 2024

PONE-D-23-24158R1 

PLOS ONE

Dear Dr. Wang, 

I'm pleased to inform you that your manuscript has been deemed suitable for publication in PLOS ONE. Congratulations! Your manuscript is now being handed over to our production team.

Kind regards, 

on behalf of

Professor Aurel Popa-Wagner 

Academic Editor

PLOS ONE